# A Combination of the Natural Molecules Gallic Acid and Carvacrol Eradicates *P. aeruginosa* and *S. aureus* Mature Biofilms

**DOI:** 10.3390/ijms23137118

**Published:** 2022-06-27

**Authors:** Maxime Gobin, Richard Proust, Stéphane Lack, Laura Duciel, Céline Des Courtils, Emmanuel Pauthe, Adeline Gand, Damien Seyer

**Affiliations:** 1ERRMECe Laboratory, 1 rue Descartes, CY Cergy Paris University, 95031 Neuville sur Oise, France; maxime.gobin@cyu.fr (M.G.); emmanuel.pauthe@cyu.fr (E.P.); adeline.gand@cyu.fr (A.G.); 2Les Laboratoires BROTHIER, 41 Rue de Neuilly, 92735 Nanterre, France; richard.proust@brothier.com (R.P.); stephane.lack@brothier.com (S.L.); laura.duciel@brothier.com (L.D.); celine.des-courtils@brothier.com (C.D.C.)

**Keywords:** biofilm, wounds, carvacrol, gallic acid, curcumin, combination therapies

## Abstract

Wound infection, especially the development of bacterial biofilms, delays wound healing and is a major public health concern. Bacteria in biofilms are more tolerant to antimicrobial agents, and new treatments to eradicate mature biofilms are needed. Combining antimicrobial molecules with different mechanisms of action is an attractive strategy to tackle the heterogeneous nature of microbial communities in biofilms. This study focused on three molecules of natural origin: gallic acid (G), carvacrol (K) and curcumin (Q). Their abilities, individually or in combination, to eradicate biofilms were quantified on mono- and dual-species mature biofilms of *Pseudomonas aeruginosa* and *Staphylococcus aureus,* the strains most commonly found in infected wounds. G presented biofilm eradicating activity on *P. aeruginosa*, whereas K had biofilm eradicating activity on *S. aureus* and *P. aeruginosa*. Q had no potent biofilm eradicating activity. The combination of G and K increased the effects previously observed on *P. aeruginosa* biofilm and led to complete eradication of *S. aureus* biofilm. This combination was also efficient in eradicating a dual-species biofilm of *S. aureus* and *P. aeruginosa*. This work demonstrates that K and G used in combination have a strong and synergistic eradicating activity on both mono- and dual-species mature biofilms of *S. aureus* and *P. aeruginosa* and may therefore represent an efficient alternative for the treatment of biofilms in wounds.

## 1. Introduction

In normal skin, commensal bacteria and immune system have a conjoint protective effect against pathogen colonization [1]. Host colonization and invasion by pathogens are facilitated by an imbalance in the skin bacterial community or skin disruption. In wounds, the presence of pathogenic bacteria organized in biofilms delays healing and is considered one of the first causes of chronic infections [2].

Studies reported evidence of a higher presence of biofilms in chronic wounds (from 20% to 90% of wounds with biofilms) than in acute wounds (6%) [3,4]. Worldwide, it is estimated that from 1% to 2% of the population in developed countries will face biofilm-associated wound infection during their lifespan. Chronic wound treatment accounts for high medical costs and represents around 2% of the health budget each year in developed countries [5].

Biofilm formation is composed of three phases: (1) attachment of planktonic bacteria to the wound surface and (2) growth and production of the extracellular matrix, maturation and (3) dispersion [6]. In a wound, bacterial biofilms can be well established within a few hours [7], being considered mature biofilms.

Current antibiotic and antiseptic treatments are powerful against planktonic bacteria but turn out to be ineffective against mature biofilms [8]. Bacteria in biofilm increase their resistance to antibacterial treatments by limiting the diffusion of antiseptics and antibiotics towards the bacteria, thanks to the secreted extracellular matrix. Their recalcitrance is also increased by the acquisition of a resistance phenotype, under quorum sensing (QS) control, including the secretion of enzymes able to degrade or neutralize antimicrobials. In addition, bacteria in the biofilm have reduced metabolic activity and are therefore not sensitive to antibiotics, the efficacy of which generally depends on growth rate. As a consequence, bacteria in mature biofilms are up to 1000-fold more resistant to antibiotics than their planktonic counterparts [8,9,10]. Therefore, new strategies are needed to fight mature biofilm infections in wounds.

There are three strategies for treating infected wounds depending on the state of the bacteria [8], which we refer to as follows: (1) antibacterial treatments, which consist of growth inhibition or the killing of planktonic bacteria, (2) antibiofilm treatments, which consist of preventing planktonic bacteria from acquiring a biofilm structure, and (3) biofilm eradicating treatments, which consist of killing bacteria already organized in a mature biofilm.

In clinical practice, the antibacterial and antibiofilm approaches are limited because wound infection is not easy to diagnose in an early stage and a mature biofilm can already be formed when infection symptoms are detected [11]. Therefore, the eradicating strategy appears to be the method of choice for biofilm treatment.

Clinical analyses showed that the critical load that may impede wound healing is approximately 10^5^ bacteria in infected wounds [12,13]. To overcome infection in wounds, the eradicating effect must thus be strong (>4 log) to eliminate the large majority of pathogenic bacteria and prevent them from reforming a biofilm. Furthermore, the eradication treatment should have a broad range of action because biofilms are generally heterogenous: composed of several species (Gram+ and Gram−) with different defense mechanisms and in different metabolic states [13]. To achieve this goal, the use of several molecules in combination seems promising, as synergistic effects were reported [13,14,15]. Despite the potential of these combination strategies, they have mainly been evaluated on the planktonic state (antibacterial effect) or on the prevention of biofilm formation.

In the present study, we looked for natural molecules that alone or in combination could have a broad and strong mature biofilm eradicating effect on Gram+ and Gram–bacteria. Active molecules of natural origin are of interest in health applications thanks to their high structural diversity and various bioactivities [16], combining, for example, antioxidant and anti-inflammatory activities with antibacterial properties. Here, we focused on the potential eradication effects on mature biofilms of three molecules: gallic acid, carvacrol and curcumin.

Gallic acid (G), carvacrol (K) and curcumin (Q) are low molecular weight natural phenolic compounds of plant origin. They differ in their formula by the presence of one or two phenol groups, the number of hydroxyl groups attached directly to the phenyl and the presence of additional chemical groups (Figure 1).

These compounds were selected for their known antibacterial properties on planktonic bacteria and their abilities to prevent biofilm formation [17,18,19,20,21,22,23,24,25]. Yet, no evidence of an eradicating activity on mature biofilm was demonstrated for G and Q. For K, an eradicating effect was previously observed only on *S. aureus* biofilm [26]. In the present study, we investigated the biofilm eradicating potential of G, K and Q, alone or in combination, against mono- and dual-species mature biofilms of *Staphylococcus aureus* (Gram+) and *Pseudomonas aeruginosa* (Gram−-), the bacterial species most commonly found in chronic wound biofilms [27,28]. Due to the appearance of antibiotic-resistant strains, they are listed by the World Health Organization as requiring highly urgent new treatments.

## 2. Results

### 2.1. Antibacterial Properties of Solutions of G, K and Q on Planktonic Bacteria

After solubilizing active molecules in water or EtOH (Appendix A), the antimicrobial effect of active molecules against *S. aureus* and *P. aeruginosa* was verified by evaluating the minimum inhibitory concentration (MIC) and minimum bactericide concentration (MBC).

To ensure that the observed effects were due to the active molecules and not to the presence of the solvent EtOH in the medium, the highest ethanol concentration enabling bacterial growth was determined and found to be 3.5% (Appendix A).

The concentrations of active molecules used to determine MIC and MBC ranged from 16.0 µg/mL to 5.0 mg/mL, depending on the solubility of the molecules and making sure it remained in conditions of final EtOH concentrations not exceeding 3.75% (*v*/*v*).

The results obtained (Table 1) showed that G, K and Q led to a complete growth inhibition at a concentration (MIC) of 2.5 mg/mL, 0.128 mg/mL and 0.064 mg/mL, respectively, on *S. aureus*, and a concentration of 2.5 mg/mL, 2.0 mg/mL and 0.128 mg/mL, respectively, on *P. aeruginosa*. For both genera, the most effective compound was Q, followed by K and G.

MBC was reached at a concentration of 5.0 mg/mL (2*MIC) and 0.512 mg/mL (4*MIC) for G and K, respectively, on *S. aureus*, and 5.0 mg/mL (2*MIC) and 2.0 mg/mL (MIC) for G and K, respectively, on *P. aeruginosa*. In the case of Q, no antibacterial effect was observed at 0.256 mg/mL, which is the maximum concentration that could be tested regarding its limited solubility and the EtOH concentration limit of 3.75%.

### 2.2. Eradicating Efficiency of Active Molecules on Mono-Species Mature Biofilms

The biofilm eradicating activity of G, K and Q was then investigated by placing the solution in direct contact with mature (24-hour-old) biofilm for 24 h, followed by bacterial biomass enumeration.

Controls performed to ensure the innocuity of the solvents on biofilms showed that 50% of water (G solvent) in a suitable bacterial medium did not affect the bacterial biomass of *S. aureus* and *P. aeruginosa* biofilms and that 7.5% EtOH (K and Q solvent, Appendix A) had a neglectable effect on biofilms with a maximum bacterial biomass reduction of 0.5 log (Figure 2).

Thus, active molecule solutions had to be diluted so that a maximum of 7.5% EtOH was reached to ensure the specificity of the observed effects. These constraints limited K and Q concentrations to 5.0 mg/mL and 0.5 mg/mL, respectively.

With a 2.0 mg/mL concentration of G, no eradicating effect was observed on both biofilms. If the concentration was increased to 5.0 mg/mL, G displayed a high eradicating effect on *P. aeruginosa* mono-species mature biofilms with a 5.8 log bacterial biomass reduction and a moderate eradicating effect on *S. aureus* with a 2.0 log reduction (Figure 2b).

A concentration of K at 0.5 mg/mL demonstrated a strong eradicating effect on *S. aureus* mature biofilms with a 6.6 log bacterial biomass reduction and a moderate effect on *P. aeruginosa* with a 1.9 log reduction. If the concentration was increased by 10 (5.0 mg/mL), the eradicating effect on *S. aureus* and *P. aeruginosa* was further enhanced with an 8.1 log and 4.2 log reduction, respectively (Figure 2b).

When biofilms were in contact with Q at 0.25 mg/mL or 0.5 mg/mL (EtOH at 7.5%), a poor effect was observed on both *S. aureus* and *P. aeruginosa* biofilms with a 1.2 log and 0.5 log bacterial biomass reduction respectively (Figure 2c). In these conditions, Q had no eradicating properties on *S. aureus* and *P. aeruginosa* mature biofilms.

These results demonstrated that G acted mainly on *P. aeruginosa* (at 5.0 mg/mL) and K on *S. aureus* (at 0.5 mg/mL) as eradicating molecules. To obtain a solution that would be efficient on *S. aureus* and *P. aeruginosa* mature biofilms, active molecules were combined (Appendix A), with the combination of K and G (0.5–5.0 mg/mL) being tested on mature biofilms. Total eradication of *S. aureus* biofilms and a strong eradicating effect with a 6.7 log biomass reduction of *P. aeruginosa* mature biofilms were observed (Figure 3).

A combination with lower concentrations of active molecules (KG 0.5–2.0 mg/mL) was also tested. In this case, the same biomass reduction was observed, showing that a lower concentration of G in combination with K led to a similar effect. G and K at concentrations of 2.0 mg/mL and 0.5 mg/mL did not have an eradicating effect on *P. aeruginosa* when used separately, but their combination led to a strong eradicating effect, suggesting a synergistic effect of the combination.

### 2.3. Eradicating Efficiency of KG Combination on Dual-Species Mature Biofilms

The KG combination was shown to have a strong eradicating effect on mono-species biofilms, but in most cases, biofilms encountered in wounds are composed of multi-bacterial species. Consequently, KG combinations were tested on dual-species mature biofilms of *S. aureus* and *P. aeruginosa* (Figure 4).

The eradicating effect was determined on the total bacterial biomass and then separately on *S. aureus* and *P. aeruginosa* biomasses. We showed that the dual-species mature biofilm was composed of an *S. aureus*: *P. aeruginosa* ratio of 2:1, which is in accordance with another study [29]. Regarding the total biomass, the 0.5–2.0 mg/mL KG combination led to a moderate eradicating effect with a 3.7 log biomass reduction. Analyzing separately the influence of the combination on *S. aureus* and *P. aeruginosa,* we observed a strong decrease in the bacterial biomass for *S. aureus* (6.2 log) and a weaker effect on *P. aeruginosa* (3.2 log). The 0.5–5.0 mg/mL KG combination was also tested and resulted in a higher eradicating effect with a decrease of 5.4 log on the total biomass and a reduction for *S. aureus* biomass of 6.3 log and for *P. aeruginosa* of 4.9 log.

The dual-species biofilm was observed by confocal scanning microscopy subsequent to live/dead staining (Figure 5).

In the absence of any treatment, biofilms appeared to be thick, dense and mainly labeled in green, suggesting the predominance of live bacteria (Figure 5a-I). With the 0.5–2.0 mg/mL KG combination, biofilms were thinner and fragmented, and red dead bacteria could be observed (Figure 5a-II). They almost completely disappeared with the 0.5–5.0 mg/mL KG combination, with a majority of red cells remaining (Figure 5a-III). The percentage of dead bacteria in dual-species biofilms with or without KG combinations was quantified (Figure 5b). Without any treatment, we observed that 40% of the bacteria were dead, and with both KG combinations, the percentage of dead bacteria reached 100%. These data are in accordance with the enumeration assays and confirm the KG combination’s direct and strong impact on dual-species biofilms.

## 3. Discussion

In this study, we confirmed the antibacterial properties of G, K and Q on planktonic cultures of *S. aureus* and *P. aeruginosa*. These properties were already described, and the effects observed in our study are mainly comparable to what was previously reported [21,24,30,31,32].

Then, we analyzed the eradicating properties of G, K and Q on 24-hour-old mature biofilms. We demonstrated a strong biofilm eradication effect by K on *S. aureus* and by K and G on *P. aeruginosa* mature biofilms. We also demonstrated a strong, synergistic eradicating effect of G and K in combination, leading to the total eradication of *P. aeruginosa* and *S. aureus* mature biofilms. Finally, a strong biofilm eradication effect of G and K in combination was also observed on the dual-species mature biofilm.

G was already described as presenting antimicrobial properties and inhibiting bacterial adherence and bacteria’s subsequent capacity to form biofilms [24]. G interferes with the QS by downregulating the synthesis of AHL (acyl homoserine lactones) [22] but also by modifying the hydrophobicity and permeability of the plasma membrane [24]. Shao et al. [33] reported an antibiofilm effect of G, preventing *Escherichia coli* and *Streptococcus mutans* biofilm formation at 8.0 mg/mL. Our results demonstrated a new property of G with an eradicating effect on *P. aeruginosa* mature biofilms with a lower concentration of 5.0 mg/mL. This eradicating effect may be due to the changes in plasma membrane properties induced by G. Indeed, in addition to altering bacterial adherence, these changes can lead to a leakage of intracellular components. It was shown that G induces a Ca^2+^ efflux from *A. naeslundii* cells [34] and a K^+^ efflux from *P. aeruginosa* [24]. Leakage of intracellular components, such as ions, can ultimately provoke cell death in numerous ways (altering gene expression, influencing cellular ion homeostasis, interfering with cell signaling, etc.). Finally, G is also able to chelate Ca^2+^ [35] and Fe^2+^ [36], which are essential metals for bacterial growth and some enzymatic activities. Therefore, G could alter the function of membrane proteins and bacterial capture iron systems such as siderophores.

Moreover, we showed a stronger effect of G on *P. aeruginosa* than on *S. aureus*. This is in agreement with the literature, as it was shown that *P. aeruginosa* was more sensitive than *S. aureus* to plasma membrane permeability and K^+^ leakage induced by G [24]. Moreover, G chelates iron and iron were previously reported to be essential for the formation of *P. aeruginosa* biofilm [36], while inversely, low iron was shown to stimulate *S. aureus* biofilm formation [37].

We demonstrated a strong eradicating activity of K on *S. aureus* and *P. aeruginosa*. A previous study [38] showed an antibiofilm effect of K, preventing *S. aureus* biofilm formation with a concentration of 0.2 mg/mL but no eradicating effect on mature biofilms. The difference can be due to the use of a lower concentration of K (0.2 mg/mL vs. 0.5 mg/mL in our study). In a different study, using different methodologies to solubilize K (Tween80), Garcia-Salinas et al. [26] observed an eradicating effect of K on *S. aureus* biofilm with a bacterial biomass reduction of 2 log at 0.5 mg/mL and of 5 log at 1.0 mg/mL. In our study, at 0.5 mg/mL of K, we obtained a stronger eradicating effect on *S. aureus* mature biofilm (6.6 log bacterial biomass reduction), and we achieved total eradication by increasing K concentration to 5 mg/mL. Moreover, we demonstrated for the first time an eradicating effect of K on *P. aeruginosa* mature biofilm (4.2 log bacterial biomass reduction at 5 mg/mL of K). The eradicating effect of K could be related to its ability to permeabilize and depolarize bacterial membranes [26]. In the Gram-negative bacterium *P. aeruginosa*, K must alter two membranes to cause the leakage of intracellular substances and, therefore, the death of the bacteria. In Gram-positive cells, only one membrane needs to be altered to cause the same damage, conferring a more efficient effect on *S. aureus*. In case of insufficient membrane defects to kill bacteria, K could also reduce the membrane potential and affect the expression of QS genes.

Due to the heterogeneity of bacteria in biofilms, the combine used of several active molecules with different antibacterial mechanisms is currently considered a necessity to efficiently eradicate biofilms [13]. Several studies have reported the importance of natural active compounds, especially when used in combination with conventional antibiotics to potentiate their effects [13,14,15]. Here, we tested the eradication efficacy of natural active compounds used alone, without the addition of conventional antibiotics. When a combination of G and K was tested on mono-species biofilm, a strong and synergistic eradicating effect was observed on *S. aureus* and *P. aeruginosa* mature biofilms. Due to its anti-adherence effect, G could destabilize biofilm, facilitating the diffusion and action of K, which permeabilizes bacterial membrane and leads to bacterial death. Consequently, the combination of these molecules not only brings an additive effect but also potentializes the activity of active molecules, leading to the combination’s synergistic eradicating properties.

On dual-species mature biofilms of *S. aureus* and *P. aeruginosa*, the eradicating effect observed with the G and K combination was reduced in comparison with the effect obtained on mono-species biofilms. In in vitro dual-species biofilms and in wounds, *S. aureus* tended to be located on the surface of the biofilm and on the edges of the wounds, whereas *P. aeruginosa* was located at the bottom of the biofilm and observed in the deeper layer of the wounds [39]. Due to this structuration, *S. aureus* is the first species to be exposed to the treatment and could serve as a protective barrier for *P. aeruginosa* bacteria. Furthermore, *S. aureus* and *P. aeruginosa* mixed biofilm is considered as an environment where the virulence of the two species is increased compared to their virulence in monoculture [29,39]. The metabolites produced by both bacteria could change their susceptibility to eradicating molecules: the siderophores produced by *P. aeruginosa* can increase the tolerance of *S. aureus* to some antimicrobials and rhamnolipid production by *P. aeruginosa* due to aminoglycoside activity may result in decreased sensitivity of *S. aureus* [40]. Polymicrobial biofilms were also described in multiple studies as more tolerant to antibiotics and able to generate synergy between species to infect and cause a wound-healing delay [28,41]. Indeed, despite the competition for nutrients between both species, *S. aureus* presence increases *P. aeruginosa* virulence. Concurrently, *P. aeruginosa* leads to the formation of small colony variants, dormant and persistent forms of *S. aureus*, which are more tolerant to stresses and treatments. That could explain the difference in efficacy observed between mono and mixed biofilms. Indeed, due to greater virulence and tolerance of bacteria, the KG combination at 0.5–2.0 mg/mL is less effective than in monoculture. By increasing the concentration of the KG combination to 0.5–5.0 mg/mL, a strong eradicating effect on mixed biofilms was obtained, with, in particular, better efficacy on *P. aeruginosa* bacteria.

## 4. Materials and Methods

### 4.1. Chemicals

Active molecules of gallic acid (G), carvacrol (K) and curcumin (Q) were purchased from Acros Organics (Fair Lawn, NJ, USA), Sigma-Aldrich (St. Louis, MO, USA) and Merck (Billerica, MA, USA), respectively.

### 4.2. Bacterial Strains and Culture Conditions

*S. aureus* CIP 4.83 and *P. aeruginosa* CIP 103.467 were purchased from the ‘Collection de l’Institut Pasteur Paris’ (CIP). *S. aureus* and *P. aeruginosa* were grown in Trypticase Soja Broth (TS) and Difco LB Broth Lennox (Becton, Dickinson Co., Franklin Lakes, NJ, USA), respectively, both overnight at 37 °C.

### 4.3. Active Molecules Solubility Assays

Active molecule solubilization was assessed in water or ethanol (EtOH—ranging from 30% to 100%) at different concentrations from 1 to 51.2 mg/mL. A manual mixing, heating (50 °C, from 5 min to 30 min) or sonication (100 W, 42 kHz, 30 min) step was performed when required. The solubility of active molecules in combination was also determined by mixing 10, 50 and 5 mg/mL of G, K and Q, respectively, in dual or triple combination in EtOH (75%).

The maximum solubility of each active molecule was defined as the concentration at which the solution appeared clear.

### 4.4. Minimum Inhibitory Concentration (MIC) and Minimum Bactericidal Concentration (MBC) Determination

MICs and MBCs were determined according to the CASFM/EUCAST (Comité de l’Antibiogramme–Société Française de Microbiologie/European Committee on Antimicrobial Susceptibility Testing) recommendations using the broth microdilution method [42]. MIC is defined as the lowest active molecule concentration at which bacterial growth completely inhibits growth in Mueller–Hinton broth, and MBC is the lowest concentration of active molecules required to kill bacteria. Controls with medium and bacteria or with medium, bacteria and solvent were made to ensure that the observed effects were only due to the active molecules and not to the solvent.

### 4.5. Mono- and Dual-Species Mature Biofilm Formation

For mono-species mature biofilm formation, bacterial overnight cultures were suspended in a suitable medium (TS or LB) at a final density of 10^6^ CFU/mL. Sterile glass coverslips with a diameter of 12 mm in 24-well plates were inoculated with the bacterial suspension (1 mL/well). The plates were then incubated at 37 °C for 24 h.

For dual-species mature biofilm formation, bacterial overnight cultures were suspended in TS medium at a final density of 10^6^ CFU/mL for *S. aureus* and 10^6^ CFU/mL for *P. aeruginosa* in the same suspension. Sterile glass coverslips in 24-well plates were incubated with the dual suspension at 37 °C for 24 h [43].

### 4.6. Biofilm Eradicating Treatments

The solutions of active molecules were diluted, individually or in combination, at the indicated final concentrations ranging from 0.1 to 5 mg/mL in a suitable medium (TS or LB). As controls, solvents used in active molecule solutions were diluted in TS or LB medium at the same ratio. The mature biofilms were incubated with the solutions for 24 h at 37 °C. Viable bacteria were then enumerated.

### 4.7. Enumeration of Viable Bacteria in Mono- or Dual-Species Biofilms after Eradicating Treatments

To enumerate bacteria [43], mature biofilms were rinsed three times with saline solution (NaCl 0.9%). The 24-well plates were then gently sonicated (100 W, 42 kHz) for 10 min with 500 µL of saline solution per well. Supernatants were collected, sonicated once again and sampled to create a series of six dilutions ranging from 1 to 10^−6^. Each dilution was spread on plate count agar for mono-species biofilm enumerations. For dual-species biofilm enumerations, dilutions were spread both on a mannitol agar plate (selective media for *S. aureus*–Becton, Dickinson and Co., Franklin Lakes, USA.) and a cetrimide agar plate (selective media for *P. aeruginosa*–VWR). After incubation at 37 °C for 24 h, colonies were counted, and the results were expressed as CFU/cm^2^ using the following relation:

Bacterial biomass = [colonies number/(dilution factor × volume spread)]/Disc surface

### 4.8. Bacterial Biomass Viability after Treatments

Biomass viability in biofilm was assessed by LIVE/DEAD staining (LIVE/DEAD *BacLight* Bacterial viability kit; Molecular Probes, Eugene, OR, USA) [44]. Images were performed using an LSM 710 confocal scanning laser microscope (Carl Zeiss, Oberkochen, Germany). Three z-stacks were acquired from random positions. The number of plates in each z-stack was adjusted to the biofilm thickness. Images were analyzed using Zen software. and Icy (https://icy.bioimageanalysis.org (accessed on 1 June 2020)) software.

### 4.9. Statistical Analysis

All the experiments were carried out at least three times independently. Results were summarized in figures and expressed as mean ± SEM. Comparisons were performed using an unpaired Student’s *t*-test, with Welch’s correction in case of unequal standard deviations. *p*-values ≤ 0.05, ≤0.01, ≤0.001 and ≤0.0001 indicated significant to highly significant differences.

## 5. Conclusions

To face the major challenge of treating biofilm infection in wounds, the use of combinations of active molecules represents a promising approach. However, the synergistic effects reported in the literature for these combination strategies are frequently evaluated on the planktonic state (antibacterial effect) or on the prevention of biofilm formation. Here, we studied the eradicating effect of three natural active molecules, used alone or in combination, on mature biofilms of two species commonly found in chronic wounds. Our results demonstrate the eradicating effect of solutions of gallic acid and carvacrol on *P. aeruginosa* and *S. aureus* mono-species mature biofilms. The combination of the two molecules induced a strong synergistic eradicating effect on both types of mono-species mature biofilms and on dual-species mature biofilms. This study confirms the potential of combinatorial strategies to eradicate mature biofilms, which opens up new promising alternatives for the treatment and cure of infected chronic wounds.

## Figures and Tables

**Figure 1 ijms-23-07118-f001:**
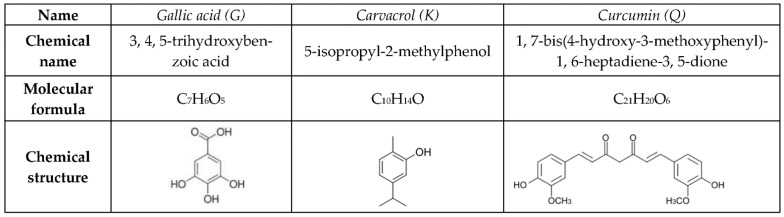
Chemical name, molecular formula and chemical structure of the three active molecules selected: gallic acid (G), carvacrol (K) and curcumin (Q).

**Figure 2 ijms-23-07118-f002:**
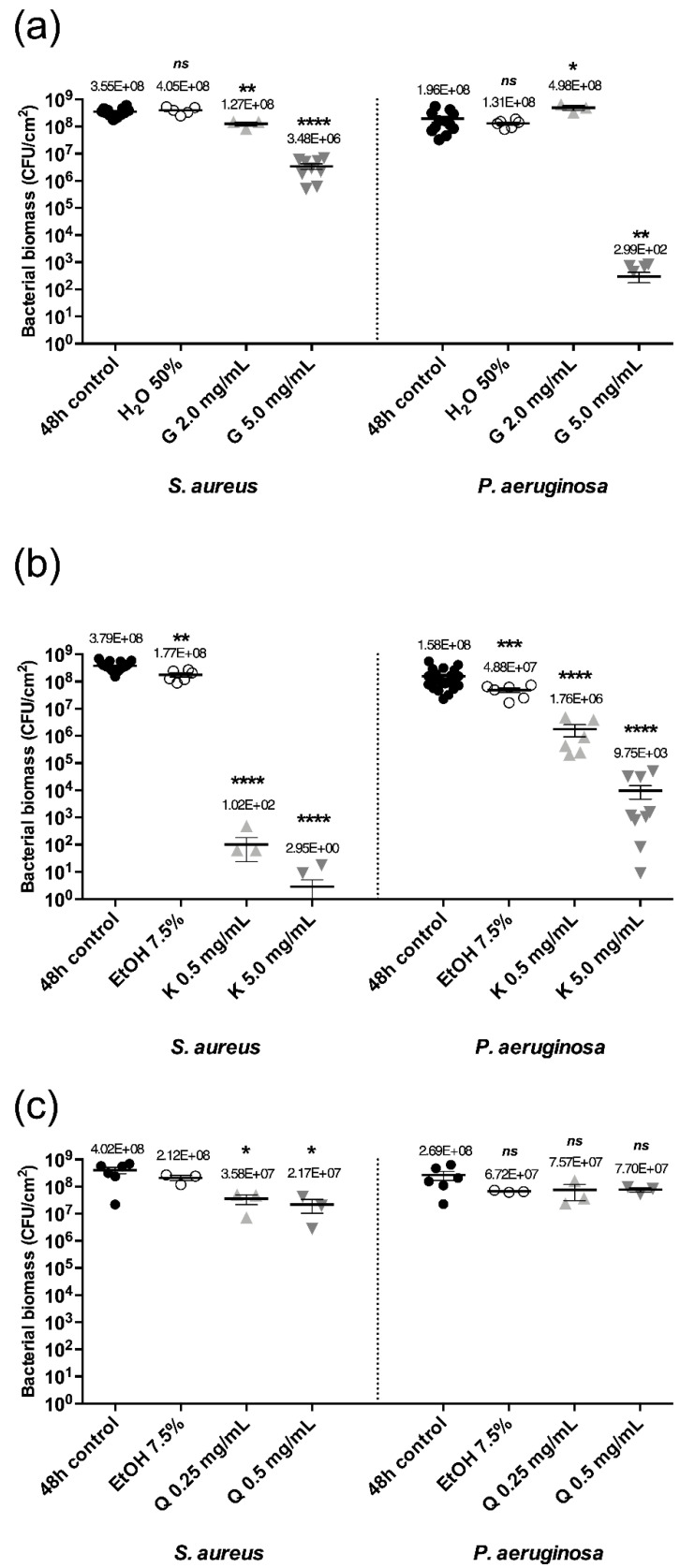
Bacterial biomass of *S. aureus* or *P. aeruginosa* mono-species mature biofilms after 24 h contact at 37 °C with solutions of G (**a**), K (**b**) and Q (**c**) at concentrations indicated. Error bars represent the standard error of the mean for at least three replicates. Statistically significant differences with 48 h control are marked with asterisks: * *p* ≤ 0.05, ** *p* ≤ 0.01, *** *p* ≤ 0.001 and **** *p* ≤ 0.0001 (ns—non-significant).

**Figure 3 ijms-23-07118-f003:**
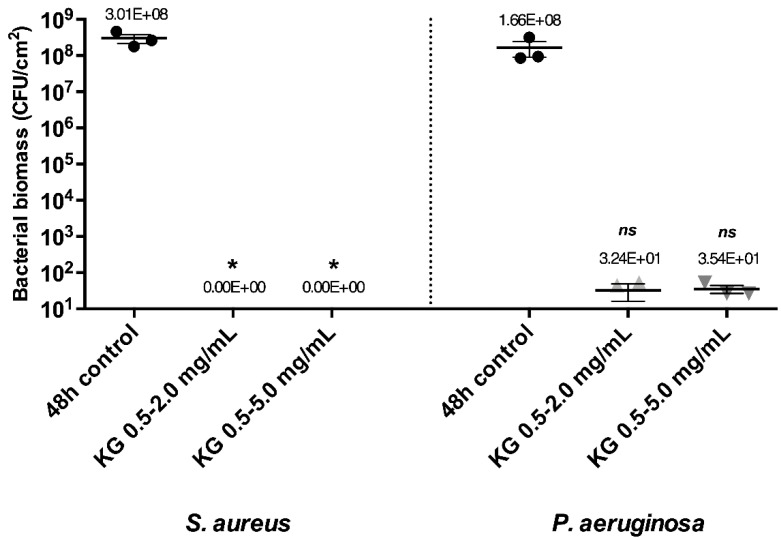
Bacterial biomass of *S. aureus* or *P. aeruginosa* mono-species mature biofilms after 24 h contact at 37 °C with solutions of the KG combination at concentrations indicated. Error bars represent the standard error of the mean for at least three replicates. Statistically significant differences with 48 h control are marked with asterisks: * *p* ≤ 0.05, (ns—non-significant).

**Figure 4 ijms-23-07118-f004:**
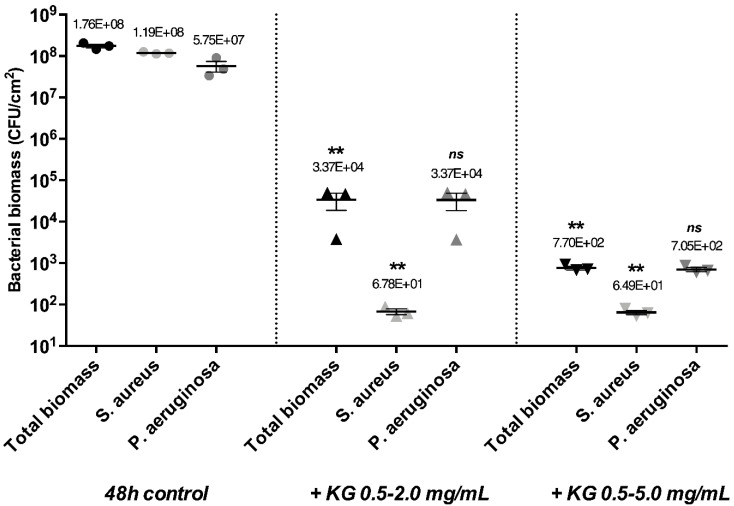
Bacterial biomass of *S. aureus*–*P. aeruginosa* dual-species mature biofilms after 24 h contact at 37 °C with KG solutions at concentrations indicated. Error bars represent the standard error of the mean for at least three replicates. Statistically significant differences with 48 h control conditions are marked with asterisks: ** *p* ≤ 0.01, (ns—non-significant).

**Figure 5 ijms-23-07118-f005:**
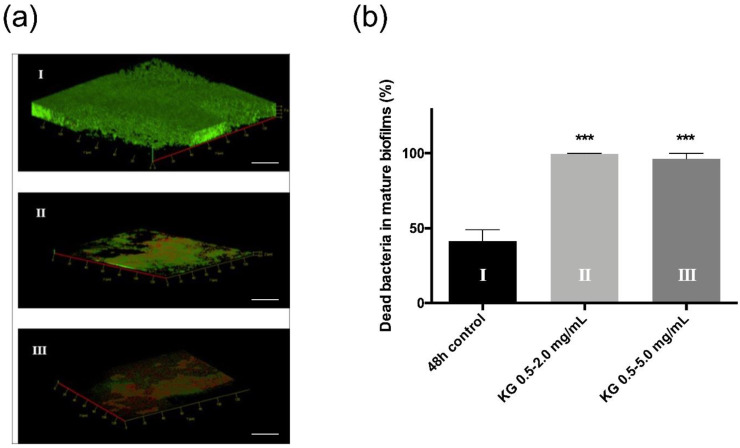
Live/dead staining images of confocal z-stacks (green = all bacteria, red = dead) of dual-species mature biofilms of *S. aureus*–*P. aeruginosa* after 24 h without active molecules (**a-I**), or in contact with KG 0.5–2.0 mg/mL (**a-II**) or KG 0.5–5.0 mg/mL (**a-III**). White bars represent 20 µm. Additionally, quantification of the percentage of dead bacteria (**b**). Statistically significant differences with 48 h control are marked with asterisks: *** *p* ≤ 0.001.

**Table 1 ijms-23-07118-t001:** Antibacterial activity of active molecules (MIC—minimum inhibitory concentration; MBC —minimum bactericide concentration).

Active Molecules			G	K	Q
**Antibacterial effect**	MIC (mg/mL)	*S. aureus CIP 4.83*	2.5	0.128	0.064
		*P. aeruginosa CIP 103 467*	2.5	2.0	0.128
	MBC (mg/mL)	*S. aureus CIP 4.83*	5.0	0.512	ND
		*P. aeruginosa CIP 103 467*	5.0	2.0	ND

ND—not determined.

## Data Availability

The data presented in this study are available on request from the corresponding author due to restrictions privacy. Publicly available datasets were analyzed in this study. This data can be found here: http://www.theses.fr/2021CYUN1069. Maxime Gobin, 2021, Doctoral thesis: Development of calcium alginate dressings with antibiofilm activity, defended on 07-12-2021.

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
