# Peer review of "A Combination of the Natural Molecules Gallic Acid and Carvacrol Eradicates P. aeruginosa and S. aureus Mature Biofilms"

_ijms, 2022, doi:10.3390/ijms23137118_

Round 1

Reviewer 1 Report

The research article "Natural active molecules combination leads to antibiofilm activity against P. aeruginosa and S. aureus matures biofilms" presents the effect of natural active molecules like gallic acid (G), carvacrol (K) and curcumin (Q), for its antibacterial and antibiofilm activities against Pseudomonas aeruginosa and Staphylococcus aureus. But, the article seems to have many serious issues associated with reported results.

The authors suggests that gallic acid (G) and carvacrol (K) in combination have strong antibiofilm activity on both mono and dual-species mature biofilms of S. aureus and P. aeruginosa. But many workers have already reported the importance of these 3 natural active molecules against several bacterial species for both antibacterial as well as antibiofilm effects! So why the authors think that this results are novel. 

Also, the introduction do not refer any significant literatures to support the rationale for this work or support the important contribution for these findings.

Line 99-101: The work presented here also cannot be considered as the effect of natural active compounds as they were extracted in ethanol solvent and its data is not presented 

Line 303-370: No references are provided for any method used or statistical analysis performed. Which also raised doubts about the results obtained.

Line 371-374: Here the authors conclude that the effects are "Biostatic" and later "Bactericidal". Which goes against their own results and discussions

Line 378-380: The authors suggest that after decades of research in natural bioactive molecules it is still a new promising opening!

All over the article the tables and figures are not provided with correct values or explainations. 

Similarly, due to the limited novelty the article still cannot be considered for publication as it lacks any in depth analysis with respect to the mechanism and/or explaination for the observed effects.

The article is too naive for publication consideration

Author Response

Response to Reviewer 1 Comments

The research article "Natural active molecules combination leads to antibiofilm activity against P. aeruginosa and S. aureus matures biofilms" presents the effect of natural active molecules like gallic acid (G), carvacrol (K) and curcumin (Q), for its antibacterial and antibiofilm activities against Pseudomonas aeruginosa and Staphylococcus aureus. But, the article seems to have many serious issues associated with reported results.

Point 1: The authors suggests that gallic acid (G) and carvacrol (K) in combination have strong antibiofilm activity on both mono and dual-species mature biofilms of S. aureus and P. aeruginosa. But many workers have already reported the importance of these 3 natural active molecules against several bacterial species for both antibacterial as well as antibiofilm effects! So why the authors think that this results are novel. 

Response 1:

We realise that one of the misunderstandings on the novelty of the work comes from the lack of precise definition of the various effects discussed here. For clarity, we have now defined the three major activities as follows:

  • Antibacterial refers to the growth inhibition or the killing of bacteria in planktonic form.
  • Antibiofilm refers to preventing planktonic bacteria from acquiring a biofilm structure
  • Biofilm eradication is now strictly restricted to the killing of bacteria already organized in a mature biofilm

Concerning previously reported activities of G, K and Q in the literature:

  • we agree with reviewer 1, that the 3 compounds have proven antibacterial activities, this was among the criteria for their selection
  • G and Q were only reported to prevent biofilm formation of some bacterial species (antibiofilm)
  • K had proven a biofilm eradication activity, only on mature biofilms of S. aureus (Garcia-Salinas and al. 2018)

In this work, we looked for molecules showing, alone or in combination, a strong (>4 Log) and broad (gram + and gram – bacteria) biofilm eradication effect.  We demonstrate for the first time:

  • using a different solubilization strategy than in Garcia-Salinas and al. a strong biofilm eradication by K on S. aureus and on P. aeruginosa mature biofilms.
  • a strong biofilm eradication activity of G on P. aeruginosa mature biofilms
  • a strong, synergistic antibiofilm effect of G and K in combination leading to the total eradication of P. aeruginosa and S. aureus mature biofilms.
  • a strong biofilm eradication effect of G and K in combination on P. aeruginosa and S. aureus dual-species mature biofilms.

The manuscript has been revised according to these definitions to clarify the results and to make sure the novelty of the work on biofilm eradication activity is understood. We thank reviewer 1 for pointing at these imprecisions, that lead us to considerably increase the clarity of the manuscript.

Point 2: Also, the introduction do not refer any significant literatures to support the rationale for this work or support the important contribution for these findings.

Response 2 : the introduction has been considerably improved to better highlight the rationale of the work and references have been added. In particular, we have added references supporting the need of efficient treatment against mature biofilms in wound, the interest of combinatorial strategies of active molecules and have clarified the previously known effects of G, K and Q on bacteria.

Point 3: Line 99-101: The work presented here also cannot be considered as the effect of natural active compounds as they were extracted in ethanol solvent and its data is not presented 

Response 3:

This first part of the paper confirms the previously known antibacterial effects (on planktonic forms) of the 3 compounds and evaluates their MIC and MIB. As requested, we have now added the antibacterial effects of ethanol in supplementary Figure 1. As shown, 3.5% ethanol has no antibacterial effect on both strains in planktonic form, and was thus used to evaluate the MIC and MBC.

Point 4: Line 303-370: No references are provided for any method used or statistical analysis performed. Which also raised doubts about the results obtained.

Response 4: We apologize for this lack of precision. References characterizing the mature biofilm model as well as the statistical methods used are now provided.

Point 5: Line 371-374: Here the authors conclude that the effects are "Biostatic" and later "Bactericidal". Which goes against their own results and discussions

Response 5: As discussed in response 1, we have now clarified the definitions of the different activities reported. Our results have two levels

  • They confirm the antibacterial activities of the compounds (growth inhibition of planktonic bacteria for G, K and, Q = bacteriostatic ; killing for G and K = bactericidal)
  • They demonstrate a strong biofilm eradication activity of G and K, when tested alone and better in combination, on mature biofilms. This second part is the novelty of the work.

The discussion and conclusion have been rewritten to better categorize those two levels and only highlight the novelty.

Point 6: Line 378-380: The authors suggest that after decades of research in natural bioactive molecules it is still a new promising opening!

Response 6: We hope the clarifications made in the manuscript following reviewer 1 criticisms have now clarified the novelty of the work and the growing interest in using combinatorial strategies against biofilms. We do not claim natural molecules are generally a promising opening but the K-G combination, for which we demonstrated a strong synergistic antibiofilm activity.

Point 7: All over the article the tables and figures are not provided with correct values or explainations. 

Response 7: The tables, figures and legends have been carefully revised and we hope all corrected. In particular, we have corrected mistakes in the MIC and MBC descriptions and in the representation of the statistics. We apologize for those that should indeed not have been missed in the previous version.  

Point 8: Similarly, due to the limited novelty the article still cannot be considered for publication as it lacks any in depth analysis with respect to the mechanism and/or explaination for the observed effects.

Response 8: The possible explanations of the observed biofilm eradication effects are now more deeply addressed in the discussion section. We thank reviewer 1 for all the constructive criticisms that have highly improved the accuracy and clarity of the paper. We believe the novelty is now well established and hope the article is suitable for publication.

Reviewer 2 Report

In the paper  Gobin et al.  "Natural active molecules combination leads to antibiofilm activity against P. aeruginosa and S. aureus matures biofilms " authors describe the antibiofilm activity of three compounds of plant origin gallic acid, carvacrol and curcumin.

General comments

In results and in conclusion should be highlited that the effect of natural ingredients on biofilm is  bacteriostatic rather than bactericidal. Not very accurately documented  conclusion that the activity of tested ingredients on formed biofilm (24 h-old) is directed on  planktonic (free-living) forms and not  biofilm (sessile) forms of bacteria.

After the digit, there should be a space before the units, e.g. not 0.25mg/mL  but 0.25 mg/mL ; not 6.6log but 6.6 log; not 40%  but 40 % ; not 30min but 30 min please check throughout the text.

In some sentences instead of ingredients compounds could be applied. The English spelling should be corrected.

Detailed comments

The title,  line 63, 86, 234:  matures biofilms should be mature biofilms

Line 15, 372 ingredients from natural origin,    ingredients of natural origin, see literature 34

Line 18,  91, 233, 234, 278, 331, 346 should be corrected in the text as it is: mono- and dual-species mature biofilms

Line 20   5.8 Log reduction CFU.cm-2 should be CFU/cm2 

Line 28 why curcumin is not mentioned in key words ?, instead of wound should be wounds

Line 36 characteristic of a forming biofilm,  English

Line 65 literature is not cited

Line 66 instead of small: low molecular weight

Line 67 instead extracted from plants: of plant origin

Line 81-84 If their  antibacterial properties and their ability to prevent biofilm formation are described among  the literature [18,21,22], no evidence of their antibiofilm properties, alone or in combination, on mature biofilms can be found.

Should be: In the papers concerning their antibacterial properties and the ability to prevent  biofilm formation [18,21,22], no evidence….

Line 90 actives ingredients solutions… should be active ingredients solutions.

Line 104, 314 concentrations not exceeding 3.75 %.... should be concentrations not exceeding 3.75 % (w/v).

Line 108 for both species,  should be for both genera

L 114 -115 (MIC: maximal inhibitory concentration; MBC: maximal bactericidal concentration). does not match the definition given earlier L 97-98 MIC (Minimum Inhibitory Concentration) and MBC (Minimum Bactericide Concentration).

Line 118 ND not detected better to write not determined

Line 149, 152  6.6log 6.6 log

Line 151 was further enhanced

Line 168, 186, 208 *P≤0.05, **P≤0.01, 168 ***P≤0.001, ****P≤0.0001**P**P≤0.01

Line 206, 207 and  quantification

 Figure 5 in the figure is  GK,  should be the same under figure capture

Line 212 and red dead bacteria and red-stained dead bacteria

Line 225 compared to our results     when comparing to our results.

L 236   Shao and al. [29]…. Shao and al. [29]

Line 249 and a reduced biofilm activity, and a reduced antibiofilm activity

Line 268, 342, 368 of 0.1 mg/mL    full stop and not a comma

Line 269 curative activity therapeutic activity

Line 273 effect is observed effect was observed

Line 288, 290 P. aeruginosa could can increase the tolerance of S. aureus  to some antimicrobials and rhamnolipid production by P. aeruginosa under due to aminoglycosides action activity could conduct to a may result in decreased sensitivity reduction of the S. aureus sensitivity [36].

Line 296 These data could explain That could explain

Line 306 and Merck (Billerica, MA, USA). and Merck (Billerica, MA, USA), respectively.

Line 310 Broth Lennox (LB – Becton, Dickinson and Co.), respectively both at…

Line 320 there was no diffraction observed through the solution., was the diffraction measured ?

Line 326 active ingredient concentration at which bacterial growth that completely inhibits growth

Line 327 MBC is the lowest concentration of ….  required to kill microorganism

Line 348 (NaCl 9%), should be (NaCl 0.9%).

In the references, the names of the bacteria should be written in italics and the name of the species in lower case not Escherichia Coli , Pseudomonas Aeruginosa but Escherichia coli, Pseudomonas aeruginosa

L 459-460 Evaluation of Nano-Curcumin Effects on Expression Levels of Virulence Genes and Biofilm Production of Multidrug-Resistant <em>Pseudo-monas</Em> <em>aeruginosa</Em> Isolated from Burn Wound Infection in Tehran, please write the title properly

Author Response

Response to Reviewer 2 Comments

General comments

In results and in conclusion should be highlited that the effect of natural ingredients on biofilm is  bacteriostatic rather than bactericidal. Not very accurately documented  conclusion that the activity of tested ingredients on formed biofilm (24 h-old) is directed on  planktonic (free-living) forms and not  biofilm (sessile) forms of bacteria.

We thank reviewer 2 for the in-depth comments on the manuscript that enabled us to highly improve it. We realise that some confusions arose from the lack of clear definition of the various effects discussed here:

  • Antibacterial refers to the growth inhibition or the killing of bacteria in planktonic form.
  • Antibiofilm refers to preventing planktonic bacteria from acquiring a biofilm structure
  • Biofilm eradication is now strictly restricted to the killing of bacteria already organized in a mature biofilm

In this work, we looked for molecules showing, alone or in combination, a strong (>4 Log) and broad (gram + and gram – bacteria) biofilm eradication effect.  We first confirm the already known antibacterial activities of G, K and Q and characterize their MIC and MBC. We then demonstrate for the first time:

  • a strong biofilm eradication by K on S. aureus and on P. aeruginosa mature biofilms
  • a strong biofilm eradication activity of G on P. aeruginosa mature biofilms
  • a strong, synergistic biofilm eradication effect of G and K in combination leading to the total eradication of P. aeruginosa and S. aureus mature biofilms.
  • a strong biofilm eradication effect of G and K in combination on P. aeruginosa and S. aureus dual-species mature biofilms.

The manuscript has been revised according to these definitions to clarify the results and to make sure the novelty of the work on antibiofilm activity is easily understood.

After the digit, there should be a space before the units, e.g. not 0.25mg/mL  but 0.25 mg/mL ; not 6.6log but 6.6 log; not 40%  but 40 % ; not 30min but 30 min please check throughout the text.

It has been corrected.

In some sentences instead of ingredients compounds could be applied. The English spelling should be corrected.

It has been harmonized all over the manuscript.

Detailed comments

The title,  line 63, 86, 234:  matures biofilms should be mature biofilms

The title has been revised.

Line 15, 372 ingredients from natural origin,    ingredients of natural origin, see literature 34

It has been corrected.

Line 18,  91, 233, 234, 278, 331, 346 should be corrected in the text as it is: mono- and dual-species mature biofilms

It has been corrected.

Line 20   5.8 Log reduction CFU.cm-2 should be CFU/cm2 

It has been corrected.

Line 28 why curcumin is not mentioned in key words ?, instead of wound should be wounds

Curcumin has been added and wounds corrected.

Line 36 characteristic of a forming biofilm,  English

It has been corrected.

Line 65 literature is not cited

It is now cited (line 116).

Line 66 instead of small: low molecular weight

It has been corrected.

Line 67 instead extracted from plants: of plant origin

It has been corrected.

Line 81-84 If their  antibacterial properties and their ability to prevent biofilm formation are described among  the literature [18,21,22], no evidence of their antibiofilm properties, alone or in combination, on mature biofilms can be found.

It has been corrected.

Should be: In the papers concerning their antibacterial properties and the ability to prevent  biofilm formation [18,21,22], no evidence….

It has been corrected.

Line 90 actives ingredients solutions… should be active ingredients solutions.

It has been corrected.

Line 104, 314 concentrations not exceeding 3.75 %.... should be concentrations not exceeding 3.75 % (w/v).

It has been corrected.

Line 108 for both species,  should be for both genera

It has been corrected.

L 114 -115 (MIC: maximal inhibitory concentration; MBC: maximal bactericidal concentration). does not match the definition given earlier L 97-98 MIC (Minimum Inhibitory Concentration) and MBC (Minimum Bactericide Concentration).

It has been corrected.

Line 118 ND not detected better to write not determined

It has been corrected.

Line 149, 152  6.6log 6.6 log

It has been corrected.

Line 151 was further enhanced

It has been corrected.

Line 168, 186, 208 *P≤0.05, **P≤0.01, 168 ***P≤0.001, ****P≤0.0001**P**P≤0.01

It has been corrected.

Line 206, 207 and  quantification

It has been corrected.

 Figure 5 in the figure is  GK,  should be the same under figure capture

It has been corrected.

Line 212 and red dead bacteria and red-stained dead bacteria

It has been corrected.

Line 225 compared to our results     when comparing to our results.

It has been corrected.

L 236   Shao and al. [29]…. Shao and al. [29]

It has been corrected.

Line 249 and a reduced biofilm activity, and a reduced antibiofilm activity

It has been corrected.

Line 268, 342, 368 of 0.1 mg/mL    full stop and not a comma

It has been corrected.

Line 269 curative activity therapeutic activity

It has been corrected.

Line 273 effect is observed effect was observed

It has been corrected.

Line 288, 290 P. aeruginosa could can increase the tolerance of S. aureus  to some antimicrobials and rhamnolipid production by P. aeruginosa under due to aminoglycosides action activity could conduct to a may result in decreased sensitivity reduction of the S. aureus sensitivity [36].

It has been corrected.

Line 296 These data could explain That could explain

It has been corrected.

Line 306 and Merck (Billerica, MA, USA). and Merck (Billerica, MA, USA), respectively.

It has been corrected.

Line 310 Broth Lennox (LB – Becton, Dickinson and Co.), respectively both at…

It has been corrected.

Line 320 there was no diffraction observed through the solution., was the diffraction measured?

No indeed, it is now corrected as follows “the concentration at which the solution appeared clear”

Line 326 active ingredient concentration at which bacterial growth that completely inhibits growth

It has been corrected.

Line 327 MBC is the lowest concentration of ….  required to kill microorganism

It has been corrected.

Line 348 (NaCl 9%), should be (NaCl 0.9%).

It has been corrected.

In the references, the names of the bacteria should be written in italics and the name of the species in lower case not Escherichia Coli , Pseudomonas Aeruginosa but Escherichia coli, Pseudomonas aeruginosa

L 459-460 Evaluation of Nano-Curcumin Effects on Expression Levels of Virulence Genes and Biofilm Production of Multidrug-Resistant <em>Pseudo-monas</Em> <em>aeruginosa</Em> Isolated from Burn Wound Infection in Tehran, please write the title properly

It has been corrected.

Reviewer 3 Report

Dear authors,

Manuscript ijms-1768110 entiteld "Natural active molecules combination leads to antibiofilm activity against P. aeruginosa and S. aureus matures biofilms" and  authored by Maxime Gobin , Richard Proust , Stephane Lack , Laura Duciel , Céline Des Courtils , Emmanuel Pauthe , Adeline Gand and Damien Seyer targeted a hot topic with a very smart and original approach. The experiments seems to be nicely designed and accurately conducted. while I am supportive to the manuscript that could be of interest for the journal readers i suggest some mandatory modifications of this manuscript to meet the journal standard:

1. The title "Natural active molecules combination leads to antibiofilm activity against P. aeruginosa and S. aureus matures biofilms" is misleading you not just tested two natural molecules and you claim that natural active  molecules combination leads to antibiofilm activity. Please be precise and choose a title that is supported by your results.

2. Keywords: there is no mention about Natural active molecules combination which is in my point of view the major finding of this paper.

3. Introduction section: please develop a part dedicated to reporting natural active molecules combination in litterature. This part have to be enriched to provide a quick overview of state of the art in the field.

4. Results: across all the results section the figures resolution and readability have to be improved. I mention particularily Figure 2, 3 and 4.

5. Discussiuon : there is no discussion of the findings of the paper and their comparison to other reports about natural active molecules combinations. Please add the relevant references in this field and discuss your results in the light of the litterature review.

6. Please add a conclusion part that is dedicated to key and main findings of the paper. This will provide journal readers an overview of the input of the paper to the field. Please delop also a part dedicated to ootlook based on the results of the study. What are the future experiments that should be conducted what are the new directions in the field.

Finally I will happy to review a new improved version of the manuscript that i could recommand for publication. 

Best regards

Author Response

Response to Reviewer 3 Comments

Dear authors,

Manuscript ijms-1768110 entiteld "Natural active molecules combination leads to antibiofilm activity against P. aeruginosa and S. aureus matures biofilms" and  authored by Maxime Gobin , Richard Proust , Stephane Lack , Laura Duciel , Céline Des Courtils , Emmanuel Pauthe , Adeline Gand and Damien Seyer targeted a hot topic with a very smart and original approach. The experiments seems to be nicely designed and accurately conducted. while I am supportive to the manuscript that could be of interest for the journal readers i suggest some mandatory modifications of this manuscript to meet the journal standard:

We thank reviewer 3 for the supportive appreciation of the work and of the importance of the topic.

  1. The title "Natural active molecules combination leads to antibiofilm activity against P. aeruginosa and S. aureus matures biofilms" is misleading you not just tested two natural molecules and you claim that natural active molecules combination leads to antibiofilm activity. Please be precise and choose a title that is supported by your results.

The title has been modified to more accurately present the obtained results: “A combination of the natural molecules gallic acid and carvacrol eradicates P. aeruginosa and S. aureus mature biofilms”.

  1. Keywords: there is no mention about Natural active molecules combination which is in my point of view the major finding of this paper.

We added the term “combination therapies” in the keywords.

  1. Introduction section: please develop a part dedicated to reporting natural active molecules combination in litterature. This part have to be enriched to provide a quick overview of state of the art in the field.

We have now added a part justifying the interest of combinatorial strategies and references.

  1. Results: across all the results section the figures resolution and readability have to be improved. I mention particularly Figure 2, 3 and 4.

We apologize for this lack of readability. We have modified the figures and hope to have solved this issue.

  1. Discussiuon : there is no discussion of the findings of the paper and their comparison to other reports about natural active molecules combinations. Please add the relevant references in this field and discuss your results in the light of the litterature review.

The discussion has been extensively revised and focused on the potential mechanisms explaining the synergistic effect of the gallic acid+carvacrol combination. References have been added accordingly.

  1. Please add a conclusion part that is dedicated to key and main findings of the paper. This will provide journal readers an overview of the input of the paper to the field. Please delop also a part dedicated to ootlook based on the results of the study. What are the future experiments that should be conducted what are the new directions in the field.

The conclusion (part 5) has been modified and focused on the novel findings of the work and on their interest for the field.

Finally I will happy to review a new improved version of the manuscript that i could recommand for publication. 

We deeply thank the reviewer for the constructive comments and hope the revised version is now suitable for publication.

Best regards

Round 2

Reviewer 1 Report

The article "Natural active molecules combination leads to antibiofilm activity against P. aeruginosa and S. aureus matures biofilms" has been considerably revised and the authors have significantly modified the article as per the critical comments. The work now seems to be more organized and is suitable for publication.

Reviewer 3 Report

Dear authors,

Many thanks for addressing all recommandations. I can now recommand youur manuscript for publication

Best regards
